



# Comparison study of COSMIC RO dry air climatologies based on average profile inversion

Julia Danzer[1/2], Marc Schwärz[1/2], Veronika Proschek[1/2], Ulrich Foelsche[2/1], and Hans Gleisner[3]

[1]Wegener Center for Climate and Global Change (WEGC), University of Graz, Graz, Austria
[2]Institute for Geophysics, Astrophysics, and Meteorology/Institute of Physics (IGAM/IP), University of Graz, Graz, Austria
[3]Danish Meteorological Institute (DMI), Copenhagen, Denmark

**Correspondence:** Julia Danzer (julia.danzer@uni-graz.at)

**Abstract.**

Recently a new approach for the production of GNSS radio occultation climatologies has been proposed. The idea is to perform the averaging of individual profiles already in bending angle space and propagating the mean bending angle profiles through the Abel transform. Climatological products of refractivity, density, pressure, and temperature are directly retrieved

from the mean bending angles.

The averaging of a large number of profiles suppresses noise in the data, enabling observed bending angle data to be used up to $80\,\mathrm{km}$ without the need of a priori information. Above that altitude some background information for the Abel integral is still necessary.

This work is a follow up study, having the focus on the comparison of the average profile inversion climatologies (API) from

the two processing centers WEGC and DMI, studying monthly COSMIC data from January to March 2011. The impact of different backgrounds above $80\,\mathrm{km}$ is tested, and different implementations of the Abel integral are investigated. Results are compared for the climatological products against ECMWF analysis, MIPAS, and SABER data.

It is shown that different implementations of the Abel integral have only little impact on the average profile inversion climatologies. On the other hand, different expansions of the bending angle profile above $80\,\mathrm{km}$ play a key role on the resulting

monthly mean refractivities above $35\,\mathrm{km}$ altitude. Below that respective altitude the API climatologies show a good agreement between the two processing centers WEGC and DMI. Due to the downward propagation within the retrieval, effects of the upper initialization lead to differences in dry temperature climatologies already at $20\,\mathrm{km}$ altitude.

Applying at both centers an exponential extrapolation to the bending angles above $80\,\mathrm{km}$, dry temperature climatologies agree between WEGC, DMI, ECMWF analysis, and MIPAS up to $35\,\mathrm{km}$ altitude within $\pm0.5\,\mathrm{K}$, and up to $40\,\mathrm{km}$ altitude

within $\pm1\,\mathrm{K}$. We conclude that up to the lower stratosphere the average profile inversion is a valid - and in computation time much faster - alternative for the production of dry atmospheric RO climatologies.



# 1 Introduction

The Global Navigation Satellite System (GNSS) Radio Occultation (RO) technique (e.g., Kursinski et al., 1997; Steiner et al., 2001; Anthes, 2011) is meanwhile accepted as valuable data source for Numerical Weather prediction (NWP) and Climate Monitoring in the Upper Troposphere and Lower Stratosphere (UTLS). Due to their high accuracy, RO data have significantly
reduced systematic errors in global weather analyses (e.g., Healy and Thépaut, 2006; Cardinali, 2009) and their potential for climate monitoring has been demonstrated with simulations studies (e.g., Leroy et al., 2006; Ringer and Healy, 2008; Foelsche et al., 2008b) and analyses (e.g., Foelsche et al., 2008a, 2009; Ho et al., 2012; Steiner et al., 2013).

NWP centers will always assimilate data that are as close as possible to the original measurement; in case of RO these are atmospheric bending angles, which can be assimilated without any bias correction. Climate monitoring based on RO data, on
the other hand, shall comprise all the atmospheric parameters down the retrieval chain, from refractivity via density and pressure to temperature, since they change differently in different parts of the atmosphere (Foelsche et al., 2008b) and temperature data are desired for comparison with data from different sources.

RO climatologies from different satellite missions like CHAMP (Challenging Minisatellite Payload) and COSMIC (Constellation Observing System for Meteorology, Ionosphere, and Climate) are very consistent (within $0.05\,\%$) up to $30\,km$ altitude
(temperature) and $35\,km$ altitude (refractivity), when the same retrieval scheme is used for all data (Foelsche et al., 2011). Data processed from different centers show differences due to structural uncertainty, which is still small at bending angle level, but increase through the retrieval chain (Ho et al., 2012; Steiner et al., 2013). The retrieval step from bending angle to refractivity is a major source for structural uncertainty, since it requires background information at high altitudes, where individual RO profiles are too noisy. When data and background are combined by statistical optimization, the observations are inversely
weighted with the measurement error. A bias in the background profile will result in a bias in the retrieved profile down to an altitude that depends on the noise of the data. The hydrostatic integral in the retrieval step from density to pressure will lead to a further downward propagation of potential biases in background data. An unbiased high altitude background - or data with low noise up to high altitudes - would therefore be highly beneficial.

Ao et al. (2012); Gleisner and Healy (2013) introduced the idea that high altitude background information could become
(largely) obsolete in climate applications, when averages over many RO profiles are used. In both studies average refractivity profiles have been obtained by averaging many COSMIC bending angle profiles in a domain and then inverting this average bending angle profile to a single refractivity profile (instead of averaging refractivity profiles, which have been obtained by inverting individual bending angle profiles). Danzer et al. (2014) have successfully applied this average profile inversion approach (API) to CHAMP data, which are more challenging due to their higher noise level. Scherllin-Pirscher et al. (2015)
introduced an alternative approach, where averaged COSMIC profiles are used to build a bending angle climatology up to high altitudes, which can then be used as background for the retrieval of individual profiles.

In this study, we test different implementations of the API approach at the Danish Meteorological Institute (DMI) and the Wegener Center for Climate and Global Change (WEGC) and validate them against independent data.



The structure of this paper is as follows: Section 2 explains the method and the different implementations at WEGC and DMI, section 3 describes the dataset, and section 4 shows result of the comparison climatologies obtained by API and ("traditionally") by averaging individual profiles obtained by IPI (individual profile inversion). In section 5 we compare the different API implementations and validate them against data from MIPAS (Michelson Interferometer for Passive Atmospheric Sounding)

and SABER (Sounding of the Atmosphere using Broadband Emission Radiometry), and against ECMWF (European Centre for Medium-RangeWeather Forecasts) analyses, followed by a summary and conclusions in section 6.

## 2    Average Profile Inversion

The retrieval step from bending angle profiles to refractivity profiles is described by an Abel transformation, which relates the refractive index $n$ to the bending angle $\alpha$:

$$\ln n(x) = \frac{1}{\pi} \int\limits_{x}^{\infty} \frac{\alpha(a)}{\sqrt{a^2 - x^2}} da \; , \tag{1}$$

where $a$ is the impact parameter and $x = nr$, with $r$ being the radius vector of a point on the ray path. The Abel integral over infinity raises a problem, since RO data are practically limited in altitude to about 80 km. Furthermore, the observed bending angle profiles suffer from a decreasing signal-to-noise ratio with increasing altitude. The need for an extrapolation step together with the handling of the noisy bending angles requires a high-altitude initialization. This is traditionally introduced at most

of the RO processing centers through a statistical optimization step (SO), where observations and background information are combined and weighted inversely with the respective errors. Different processing centers use different kinds of background information (e.g. from climatological models such as MSIS (Mass Spectrometer and Incoherent Scatter Radar), or meteorological data such as ECMWF analysis) and different implementations of the statistical optimization step (e.g., Gorbunov, 2002; Gobiet and Kirchengast, 2004; Lohmann, 2005).

The basic idea of the API approach is that averaging of the data in bending angle space suppresses the noise in the data, so that the observed bending angle can be used up to 80 km and the SO step becomes largely obsolete. Above $80 \, \mathrm{km}$ some kind of background information is still necessary.

The main steps of the average profile inversion retrieval can be summarized as a) generation of the average bending angle as a function of impact altitude, b) change of height variable from impact altitude to impact parameter, $a$, using an average

radius of curvature, $\overline{R}_c$, c) expansion of the average bending angle profiles to infinity, which we introduce as "high altitude expansion", d) retrieval of the average refractivity as a function of $x = nr$ using the Abel transform (Eq. 1), and e) change of height variable to mean-sea level altitude, using the same radius of curvature as in step b).

### 2.1    WEGC implementation

The latest implementation of the inversion of the individual profiles at WEGC is currently in an experimental state. It is based

on the so-called base-band method (Kirchengast et al., 8-14 September 2016, 2017). As input data, excess phase profiles





provided by the COSMIC Data Analysis and Archiving Center (CDAAC) of the University Corporation for Atmospheric Research (UCAR), Boulder, Colorado were used. From these data bending angle profiles are calculated by applying a combined geometric optics (see Appendix A in Schwarz et al. (2017a)) and wave optics (Gorbunov and Lauritsen, 2004; Gorbunov and Kirchengast, 2018) bending angle retrieval. To obtain ionosphere-free bending angles, the method of Sokolovskiy et al. (2009)

is applied on the calculated bending angles. After that each bending angle profile is statistically optimized using an ECMWF short-range bias corrected forecast as background profile (Li et al., 2013, 2015). As a next step the refractivity is calculated, applying the method described in Appendix B in Syndergaard and Kirchengast (2016) on the residual state. Dry pressure and dry temperature are obtained by evaluating the hydrostatic integration (once more on the residual state, c.f., Appendix A in Schwarz et al. (2017b)). The monthly climatologies are then obtained by averaging the individual profiles into latitude bins.

The API processing at WEGC follows the basic description of Sec.2. The mean, median, and so-called medmean bending angle climatologies are calculated. Medmean uses mean bending angle values up to $50\,\mathrm{km}$, median values above $60\,\mathrm{km}$, and a linear combination inbetween (Gleisner and Healy, 2013). Together with the average bending angles, the average radii of curvature are built, where we test three different implementations of mean $\overline{R}_c$ (see Eq. 2 - Eq. 4, and Fig. 1). The first formulation of $\overline{R}_c$ follows Gleisner and Healy (2013), and is determined as a sum of all single radii of curvature per bin ($R_{c,i}$

with occultation $i$), divided by the number of occultations $m$ in a bin:

$$\overline{R}_c = \frac{1}{m} \sum_{i=1}^{m} R_{c,i} \ . \tag{2}$$

As an alternative formulation we test the Earth's mean radius of curvature at latitude $\varphi$:

$$\overline{R}_c = \frac{2}{\frac{1}{\mathrm{M}} + \frac{1}{\mathrm{N}}} \ , \tag{3}$$

with $\mathrm{M}(\varphi) = \frac{\mathrm{ab}^2}{((\mathrm{a}\cdot\cos\varphi)^2 + (\mathrm{b}\cdot\sin\varphi)^2)^{3/2}}$, $\mathrm{N}(\varphi) = \frac{\mathrm{a}^2}{\sqrt{(\mathrm{a}\cdot\cos\varphi)^2 + (\mathrm{b}\cdot\sin\varphi)^2}}$, a is the Earth's equatorial radius of $6378.1370\,\mathrm{km}$, and b

is the Earth's polar radius of $6356.7523\,\mathrm{km}$ (WGS84, World Geodetic System 1984). Furthermore we study the formulation of Earth's Gaussian radius of curvature at latitude $\varphi$ (Torge, 2001):

$$\overline{R}_c = \frac{\mathrm{a}^2\mathrm{b}}{(\mathrm{a}\cdot\cos\varphi)^2 + (\mathrm{b}\cdot\sin\varphi)^2} \ . \tag{4}$$

The l.h.s. of Fig. 1 compares the mean radius of curvature, using the three different formulations of $\overline{R}_c$ (Eq. 2 to Eq. 4), studying monthly $5°$-zonal COSMIC data from January 2011. Obviously, the Mean $\overline{R}_c$ (green line) and the Gaussian $\overline{R}_c$ (red dashed

line) show almost no differences (Eq. 3 and Eq. 4, respectively). Compared to the average $\overline{R}_c$ per bin (Eq. 2, AvProf - blue line) differences increase in the tropics between about $2\,\mathrm{km}$ and $8\,\mathrm{km}$. Studying the impact of those differences on resulting dry temperature climatologies (r.h.s. of Fig. 1), only negligible implications are found. The different implementations of $\overline{R}_c$ lead to variations of $1/1000\,\mathrm{K}$ to about a few $1/100\,\mathrm{K}$ up to $80\,\mathrm{km}$ altitude, comparing the same monthly $5°$-zonal COSMIC climatology.

For evaluating the Abel integral, different methods for the upper boundary values at $80\,\mathrm{km}$ have been tested (high altitude expansion). Initially we study monthly means of ECMWF analysis fields converted to refractivity, as value for the Abel integral




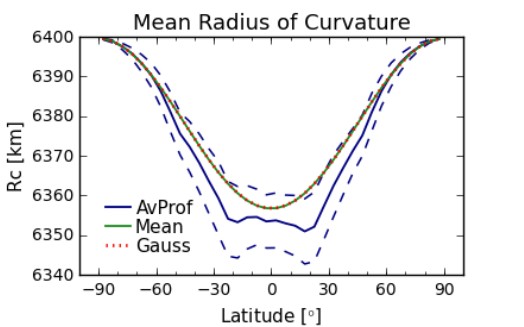
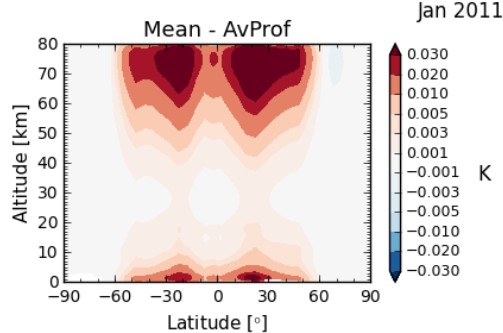

**Figure 1.** l.h.s.: Comparing different implementations of the mean radius of curvature $\overline{R}_c$, analyzed for a $5°$-zonal COSMIC climatology from January 2011.

r.h.s.: Dry temperature difference, comparing the implementation of $\overline{R}_c$ using Eq. 3 (Mean) to Eq. 2 (AvProf).

from infinity to $80\,\mathrm{km}$ (Kirchengast et al., 2017). The data sets are labeled as "fulltop". As an alternative also an exponential extrapolation of the bending angles to infinity is tested (exptop), where scale height and fitting coefficient are calculated from a log-linear fit to each average bending angle profile. Furthermore, the case of setting the average bending angles to zero above $80\,\mathrm{km}$ is studied (notop). Additionally a sensitivity study from the fulltop value to notop in 1/5 incremental steps is performed

(notop, top1, top2, top3, top4, fulltop). For an overview of all data sets see Tab. 1.

Finally the average bending angles are forwarded through the Abel integral using the base-band method, and are further processed as described for the individual profile processing at WEGC.

## 2.2 DMI implementation

The DMI data based on the standard IPI processing were obtained from a reprocessed climate data record provided by the

ROM SAF, which is a decentralized RO Satellite Application Facility under EUMETSAT. The COSMIC data in this data set are based on input data from the CDAAC archive at UCAR. The individual bending angle profiles are calculated using a combination of geometric optics and wave optics approaches, followed by smoothing and merging with a background profile taken from the BAROCLIM climatology (Scherllin-Pirscher, 2013; Scherllin-Pirscher et al., 2015). The statistical-optimization step is followed by an inverse Abel transform, to retrieve the refractivity profile, and a hydrostatic integration to retrieve the

dry-temperature profile (Lauritsen et al., 2011). The monthly climatologies are then obtained by averaging the invidual profiles into latitude bins.

The API processing used by DMI in the present study is described in more detail in Gleisner and Healy (2013). The average bending-angle profiles are computed as a combination of mean (up to $50\,\mathrm{km}$), median (above $60\,\mathrm{km}$), and a linear combination of the two (from $50\,\mathrm{km}$ to $60\,\mathrm{km}$). The statistical analysis is done on a common impact altitude grid, which is mapped to an

impact parameter grid using an average radius of curvature, $\overline{R}_c$, according to Eq. 2. This is followed by an extension of the average bending angle profile from the top of the profile up to infinity assuming a constant scale height of $7.5\,\mathrm{km}$, in contrast



to WEGC, which calculates the scale height individually for each mean bending angle. The exponential extrapolation of the bending angles is called "exptop" in the data sets. The Abel transform (Eq. 1) is then used to retrieve refractivity as function of $x = nr$, which is mapped to mean-sea level altitude, $H$, using the mean radius of curvature, $\overline{R}_c$.

In the present study, DMI used an implementation of the inverse Abel transform provided by the ROM SAF ROPP software package (Culverwell et al., 2015). This assumes that the bending angle, $\alpha$, can be approximated as a linear function of impact parameter, $a$, between successive grid points. The sub-integrals between the grid points can then be solved analytically, and the refractivity at a certain height, $x$, is simply given by a sum of the contributions from the atmospheric layers from height $x$ to the top of the atmosphere.

## 3   Data sets

We analyze occultations from the six-satellite mission FORMOSAT-3/COSMIC for the year 2011, from January until March. Excess phase profiles and precise orbit information were retrieved from the UCAR/CDAAC database and then further converted into bending angle profiles and dry air profiles (Level L2a processing) at the WEGC and also at the DMI, using the rOPS-ex (reference Occultation Processing System-experimental) and ROPP version 8 (the ROM SAF Radio Occultation Processing Package), respectively. The processing chain from a single bending angle profile down to dry temperature we introduce as "individual profile inversion" (IPI). In a next step the profiles were binned into monthly $5°$-zonal climatologies (IPI climatologies) at both processing centers.

Furthermore, using the same COSMIC satellite data sets, average profile inversion climatologies (API climatologies) were produced, as described in Sec. 2. API climatologies are available from bending angle down to dry temperature (L2a processing) on a monthly $5°$-zonal grid. At WEGC, the API climatologies were produced using processing routines from rOPS-ex (Abel inversion, hydrostatic integral), at the DMI, processing routines were used from the ROPP, respectively. We tested in the API processing different high altitude expansions (see description in Sec. 2). An overview of the data sets and all data versions (fulltop, exptop, etc.) is given in Tab. 1. For clarification of the different data versions and their notations we give two examples:

The label "WEGC (L1b DMI) - fulltop" refers to an input bending angle climatology generated at the DMI (Gleisner and Healy, 2013), hence "L1b DMI", and then forwarded through processing routines from WEGC, using the WEGC high altitude expansion "fulltop". On the other hand "DMI (L1b DMI) - exptop" uses the same bending angle input climatology from the DMI, and forwards the climatology through DMI processing routines, using the "exptop" high altitude expansion. So basically those two processing versions share the same input bending angle climatology, but differ in the further processing (WEGC and DMI) and their handling of the top (fulltop and exptop).

As reference data sets co-located ECMWF (European Centre for Medium-Range Weather Forecasts) profiles from ECMWF analysis data were studied on $5°$ latitudinal bins. The analysis data fields were used in a T42L91 resolution, since the T42 horizontal resolution matches the resolution of RO data ($\sim 300\,\mathrm{km}$). The ECMWF analysis climatologies were used as reference data sets from bending angle down to temperature (i.e., Level L2a climatologies), see Tab. 2.



Furthermore we use data from the MIPAS (Michelson Interferometer for Passive Atmospheric Sounding) and SABER (Sounding of the Atmosphere using Broadband Emission Radiometry) instruments as reference data sets to RO climatologies. The MIPAS instrument, onboard ENVISAT (Environmental Satellite), operated from July 2002 until April 2012, providing global temperature, pressure, and trace gas observations in an altitude range from about $6\,\mathrm{km}$ to $70\,\mathrm{km}$. SABER, onboard the

5 TIMED (Thermosphere Ionosphere Mesosphere Energetics and Dynamics) satellite, measures data since 2001, providing temperature, pressure, density, geopotential height, and trace species. The coverage is nearly global between 52°S - 82°N and 82°S - 52°N, respectively alternating every two months, providing a continuous coverage from 52°S - 52°N, in an altitude range from about $10\,\mathrm{km}$ to $180\,\mathrm{km}$. A validation study of MIPAS temperature in the middle atmosphere showed good agreement to SABER temperature ($< 0.5\,\mathrm{K}$) in mid-latitude in the upper troposphere (García-Comas et al., 2012).

At WEGC a master thesis has been conducted, performing a profile to profile inter-comparison study between WEGC RO OPSv5.6 data (Schwärz et al., 2016) and ECMWF, MIPAS, and SABER data (Innerkofler, 2015). The study shows good agreement between ECMWF analysis and RO data up to $80\,\mathrm{km}$, with temperature differences of about $\pm 1\,\mathrm{K}$. MIPAS data also show good agreement up to $40\,\mathrm{km}$ altitude with differences of about $\pm 1\,\mathrm{K}$, between $40\,\mathrm{km}$ to $50\,\mathrm{km}$ height, differences increase to about $\pm 2\,\mathrm{K}$. In contrast to MIPAS, SABER data show a cold bias of $3\,\mathrm{K}$ between $20\,\mathrm{km}$ to $35\,\mathrm{km}$. From $35\,\mathrm{km}$ to

$45\,\mathrm{km}$ altitude differences decrease to $\pm 2\,\mathrm{K}$.

**Table 1.** Data sets from the COSMIC mission, studying always monthly 5°-zonal climatologies of the dry atmosphere.

| Date | Processing | Inversion | L1b Bending Angle Climatology/Profiles | Parameters | Expansion | Label |
|---|---|---|---|---|---|---|
| 01-03 2011 | rOPS-ex | API, IPI | L1b WEGC-ex | $\alpha$, N, $\rho$, p, T | fulltop | WEGC (L1b WEGC) |
| 01-03 2011 | ROPP | API, IPI | L1b DMI | $\alpha$, N, T | exptop | DMI (L1b DMI) |
| 01-03 2011 | rOPS-ex | API | L1b DMI | $\alpha$, N, $\rho$, p,T | fulltop | WEGC (L1b DMI) |
| 01-03 2011 | rOPS-ex | API | L1b DMI | $\alpha$, N, $\rho$, p,T | exptop | WEGC (L1b DMI) |
| 01-03 2011 | rOPS-ex | API | L1b DMI | $\alpha$, N, $\rho$, p,T | notop | WEGC (L1b DMI) |
| 01 2011 | rOPS-ex | API | L1b DMI | $\alpha$, N, $\rho$, p,T | top1, top2, top3, top4 | WEGC (L1b DMI) |
| 01-03 2011 | ROPP | API | L1b DMI | $\alpha$, N, T | notop | DMI (L1b DMI) |

**Table 2.** Reference data sets to Tab. 1, studying monthly 5°-zonal climatologies.

| Date | Reference Data | Version | Vertical Range | Parameters | Global Sampling | Label |
|---|---|---|---|---|---|---|
| 01-03 2011 | ECMWF analysis | T42L91 | 91 model levels | N, $\rho$, p, T | 4 times/day | ECMWF |
| 01-03 2011 | MIPAS data | ML2PPv7.03 | 6 km - 80 km ~3 km resolution | p, T | ~800 profiles/day | MIPAS |
| 01-03 2011 | SABER data | GATSv2.05 | 10 km - 80 km ~2 km resolution | $\rho$, p, T | ~1500 profiles/day | SABER |



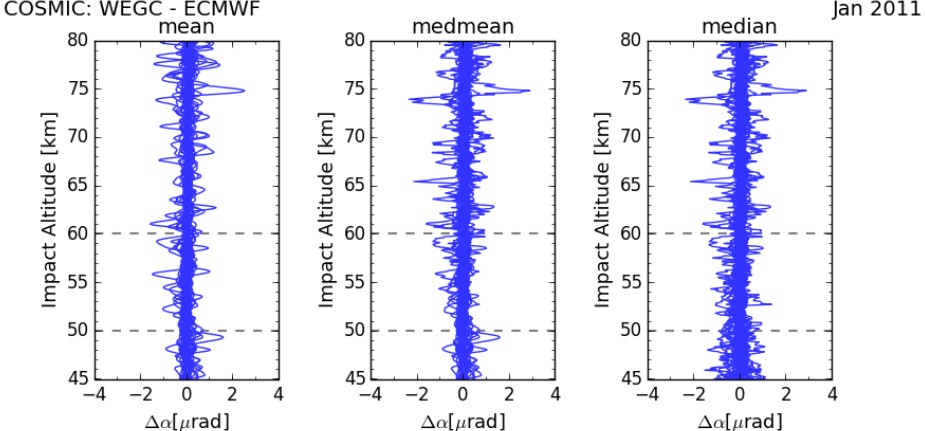

**Figure 2.** L1b WEGC: Bending angle difference of mean, medmean, and median relative to ECMWF analysis, analyzing January 2011.

## 4 WEGC API climatologies

We start our analysis with the investigation of API climatologies from WEGC. API climatologies have already been thoroughly tested at the DMI (Gleisner and Healy, 2013; Danzer et al., 2014), showing very good agreements between API and IPI refractivity climatologies up to 35 km altitude.

Initially, we investigate monthly 5°-zonal rOPS-ex bending angle climatologies (WEGC L1b) for the COSMIC satellite mission and January 2011. Fig. 2 shows the difference of the mean, medmean, and median bending angles relative to co-located ECMWF analysis. Obviously the bending angles show strong variations relative to ECMWF analysis. We emphasize that those bending angles are only recently generated experimental data, which is one reason why we later on continue our analysis based on DMI bending angles.

As a next step we compare API to IPI climatologies, using the rops-ex bending angles (WEGC L1b) as input for the API and IPI processing. Fig. 3 shows the difference between API and IPI refractivity (left column) and dry temperature (right column) climatologies, from January to March 2011 (top to bottom). Analyzing refractivity differences, API and IPI climatologies show almost identical results up to 40 km altitude. Only around the tropopause and in the height range between 40 km to 50 km altitude differences vary between about 0.2 % and 0.6 %. This confirms the result from previous studies that the average

profile inversion is a valid alternative to the individual profile inversion, since no significant differences are introduced.

Continuing the processing down to dry temperature and studying the differences between the two approaches, the API and IPI climatologies agree within the RO core region of 35 km altitude (lower stratosphere) very well. Above that height, differences start to increase with about 1 K every 3 km to 5 km altitude.

Summarizing the main results from this analysis: First, it was possible to successfully implement the API approach at the

WEGC, as it has been done in previous studies at the DMI. Second, the API approach does not introduce major differences within the RO core region of 35 km. Hence, it is a valid alternative for climate analysis in the lower stratosphere.




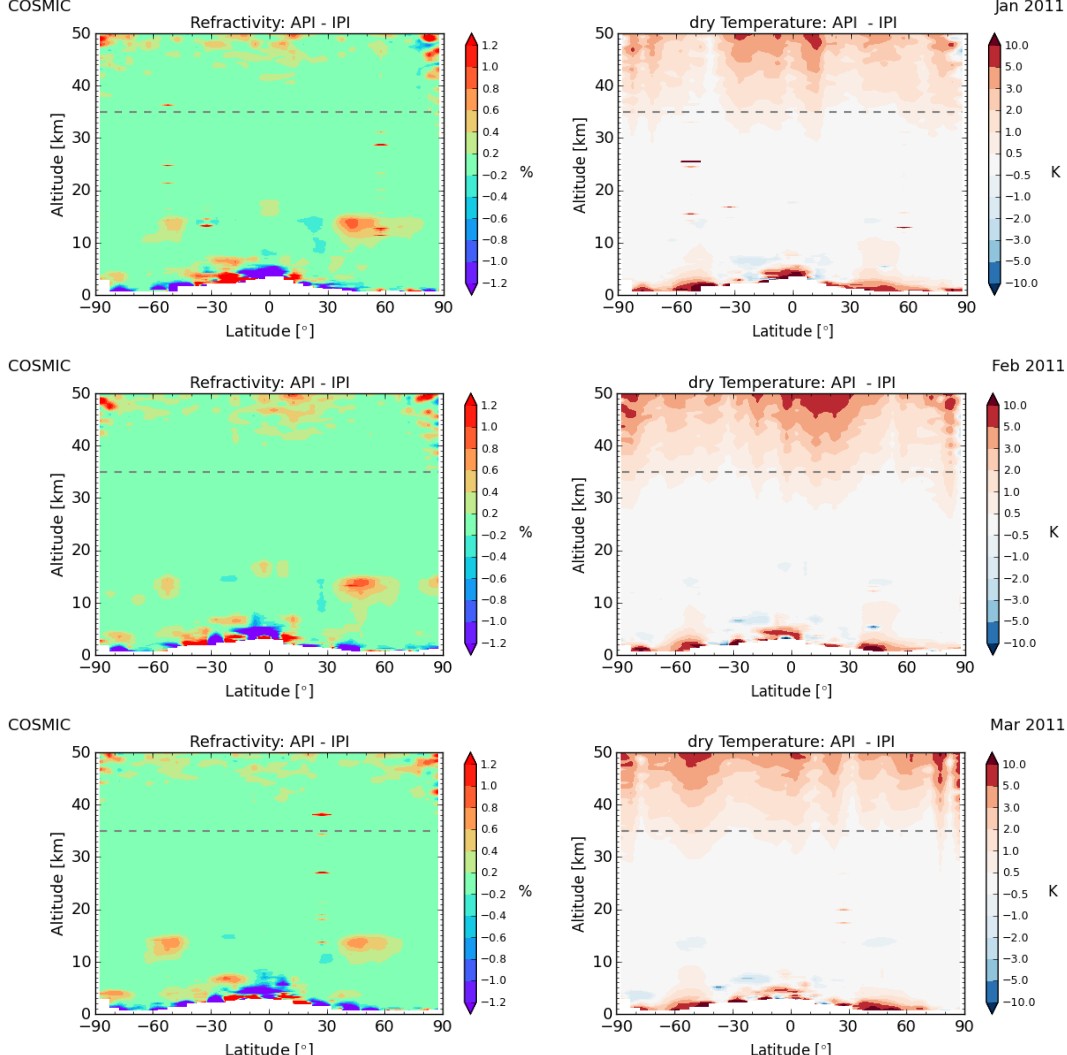

**Figure 3.** WEGC: Difference between average profile inversion (API) and individual profile inversion (IPI), analyzed for refractivity (left column) and dry temperature (right column), using L1b WEGC bending angles as input, studied from January to March 2011 (top to bottom).

## 5 Comparison of WEGC and DMI API climatologies

The main focus of this study is a thorough comparison of API climatologies between the two processing centers WEGC and DMI. Since we want to understand how differences enter in the processing from API bending angle climatologies to refractivity climatologies, we decided to always use the same input bending angle climatology for both processing systems. For practical reasons we chose to study bending angle climatologies from the DMI, labeled as DMI L1b, since WEGC rOPS-ex is still in the development process (see Fig. 2). Fig. 4 shows the monthly 5°-zonal mean, medmean, and median bending angle climatologies



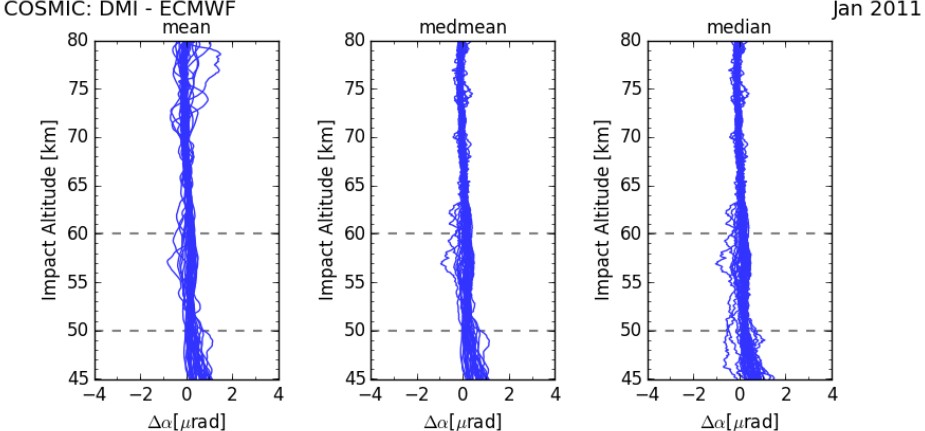

**Figure 4.** DMI: Bending angle difference of mean, medmean, and median relative to ECMWF analysis, studying Jan 2011.

relative to ECMWF analysis for January 2011. February and March 2011 show a very similar behavior, hence we only present results for one month.

In the Abel integral we use as estimate for the central bending angle value per bin the mean-median combination called medmean, since at higher altitudes the mean value suffers from large-scale wiggles and the median becomes a more robust
estimate (see discussions in Gleisner and Healy (2013), Danzer et al. (2014)).

## 5.1   API refractivity climatologies

In this section we show a first comparison of API refractivity climatologies between processing centers, i.e., WEGC and DMI. In Fig. 5 we investigate the difference of API refractivity climatologies relative to co-located ECMWF analysis, from January until March 2011. The left column corresponds to WEGC processing, while the right column corresponds to the
DMI processing routines. What strikes out in this plot series is that the results at WEGC above $35\,\mathrm{km}$ (left column) are always much larger relative to ECMWF, compared to the DMI results (right column). Below $35\,\mathrm{km}$ results are in general very consistent between WEGC and DMI, however in the tropopause region the data show again a slight increase relative to ECMWF and compared to the DMI. Since both processing centers are using the same input bending angle climatologies (DMI L1b) differences can only enter through alternativ handling of the top (fulltop and exptop) and also different implementations
of the Abel integral.

In order to illustrate the discrepancies between WEGC and DMI more strongly, Fig. 6 studies directly the differences between the two API climatologies. Clearly the plot series confirms for all months that WEGC and DMI processing are almost identical up to $35\,\mathrm{km}$ altitude, only in the tropopause region we find differences of about $0.2\,\%$.

Nevertheless, we want to understand the occurring differences between WEGC and DMI, which is why we try to separate
the underlying factors in the next two sections, i.e, the high altitude expansion and the Abel integral.



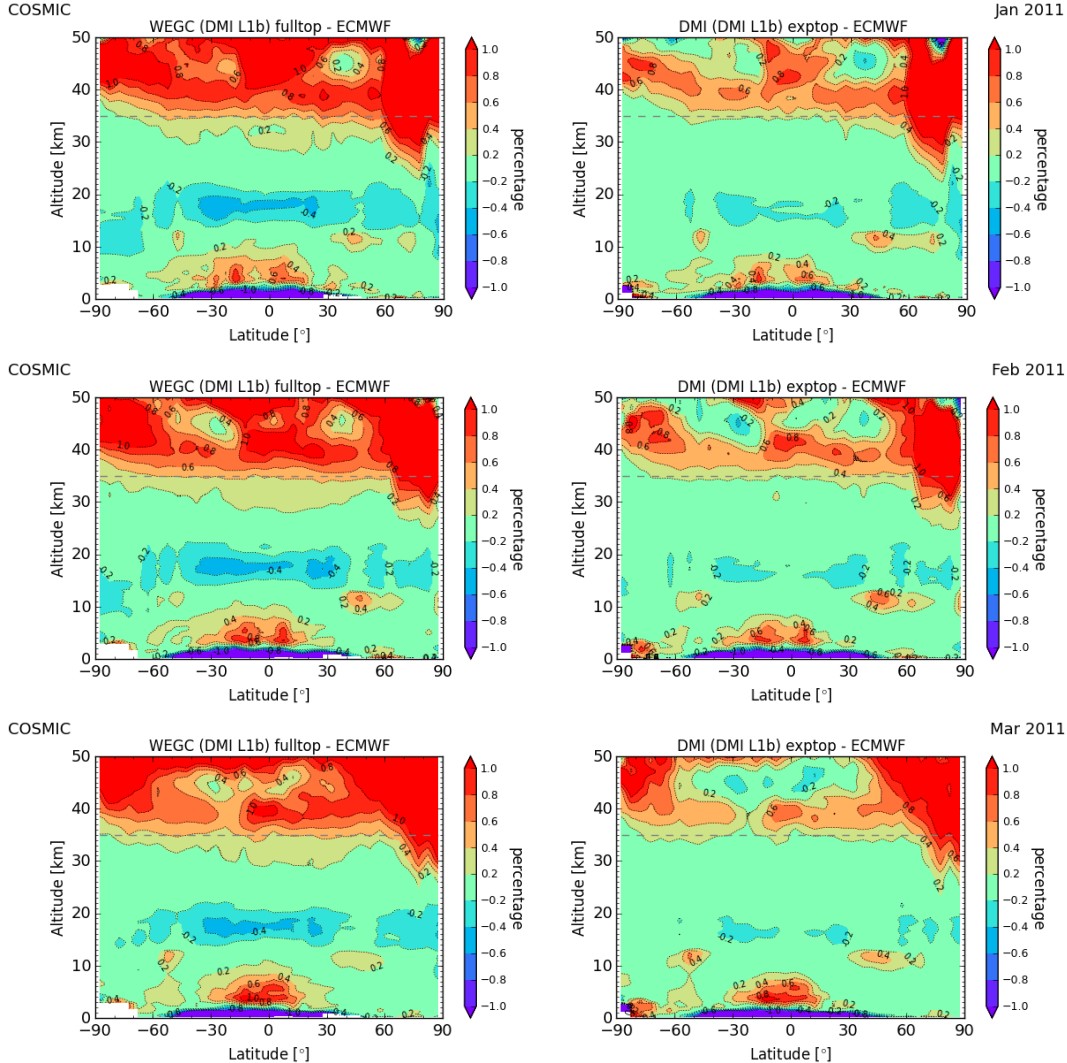

**Figure 5.** API refractivity climatologies relative to ECMWF analysis, comparing WEGC processing (left column) to DMI processing (right column), from Jan 2011 to Mar 2011, using the same bending angle profiles as input (DMI L1b).

## 5.2 Testing the impact of the Abel integral

In Fig. 7 the sole influence of different implementations of the Abel integral is investigated, exemplary shown on January 2011. To realize that we switch off the high altitude expansion at both processing centers and set the bending angle climatologies to zero above $80\,\mathrm{km}$ (top row, notop). Furthermore, as a test, we initialize the bending angles at both centers with an exponential extrapolation (bottom row, exptop).







**Figure 6.** API refractivity climatologies difference between WEGC and DMI processing from Jan 2011 to Mar 2011, using the same bending angle profiles as input (DMI L1b).

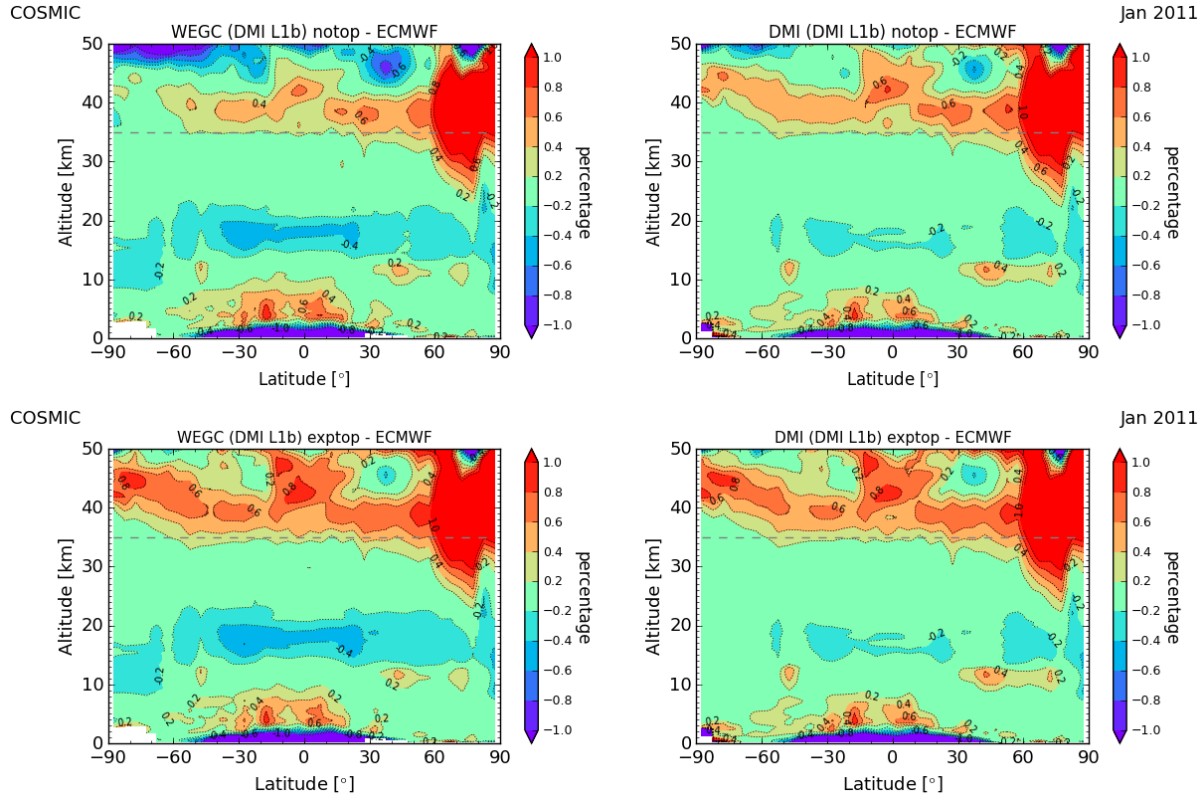

**Figure 7.** API refractivity climatologies relative to ECMWF analysis, comparing WEGC processing (left) to DMI processing (right), studying notop (first row) and exptop (second row), exemplary for Jan 2011.

Obviously, results become immediately very similar between the two processing centers WEGC and DMI, once the high altitude expansion is handled in the same way. Notop (first row), as well as, exptop (second row) agree very well between WEGC and DMI, even above 35 km altitude. Only in the region around the tropopause small differences exist.





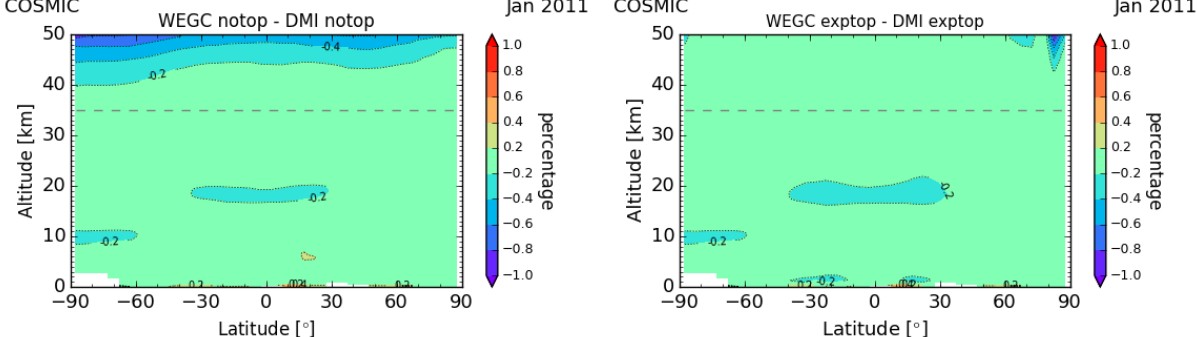

**Figure 8.** Difference between WEGC and DMI API refractivity climatologies, studying notop (left plot) and exptop (right plot), exemplary for Jan 2011.

Once again, discrepancies are more clearly illustrated by studying differences directly between WEGC and DMI (Fig. 8). We find the already noticed $0.2\%$ differences in the tropopause region. Furthermore we see that differences start to increase above $40\,\mathrm{km}$ with $0.2\%$ for the notop case (left plot). Almost identical results are found up to $50\,\mathrm{km}$ altitude for the exptop case (right plot), with small exceptions in the high altitude polar north. Since integration starts at $80\,\mathrm{km}$ altitude only in the notop case, absolute values at $50\,\mathrm{km}$ are smaller than for the exptop case, and the same absolute difference corresponds to a higher relative difference. Hence differences begin to increase at already $40\,\mathrm{km}$ altitude for notop.

To sum up, these results suggest that the handling of the top has a significant influence on the API refractivity climatologies above $35\,\mathrm{km}$. On the other hand, different implementations and discretizations of the Abel integral seem to lead to only small differences, mainly in the tropopause region. Hence we conclude that in the context of the average profile inversion approach, a major focus should be laid on the handling of the bending angle profiles above $80\,\mathrm{km}$.

### 5.3 Testing the impact of different high altitude expansions

This section presents a first attempt to address the high altitude expansion. From the initial testing of the WEGC and DMI API processing it is clear that the choice of the top has a substantial effect on the resulting refractivity climatologies above $35\,\mathrm{km}$. The question of how to handle the top of the bending angles is of course a general question, also in respect to individual profile processing. The rOPS-ex of WEGC is still in the development process, where at the moment a lot of effort is put into answering that question.

In a first analysis we investigate the sensitivity of the API refractivity climatologies with respect to different top values, for January 2011. We start in Fig. 9 from a top value of zero and increase the top value in 1/5 incremental steps, until reaching the fulltop value of the rOPS-ex. Clearly, the results are insensitive to different top values below $35\,\mathrm{km}$, while errors increase at $40\,\mathrm{km}$ already up to $1\%$ relative to ECMWF analysis for the fulltop value.

Figure 10a investigates the difference of single API refractivity climatologies relative to ECMWF analysis for six example zonal bins up to $50\,\mathrm{km}$, comparing the above shown different choices of the top value. Obviously the notop choice usually





**Figure 9.** Sensitivity of API refractivity climatologies relative to different handling of the high altitude expansion, analyzed against ECMWF analysis. The plots start from a top value of zero (notop) and increase in 1/5 incremental steps (top1, top2, top3, top4) to the full value (fulltop).

agrees better with ECMWF, while the fulltop value shows largest differences within slightly above $1\,\%$ at $50\,\mathrm{km}$. Differences between the varying choices of the top start to increase at the usual characteristic altitude of $35\,\mathrm{km}$, with a maximal spread between notop and fulltop of about $1\,\%$ at $50\,\mathrm{km}$ altitude. Only in northern high latitudes differences are larger relative to ECMWF analysis, which could be related to different sampling of the upper stratosphere lower mesosphere (USLM) distur-
5   bance in January 2011 (Greer et al., 2013).



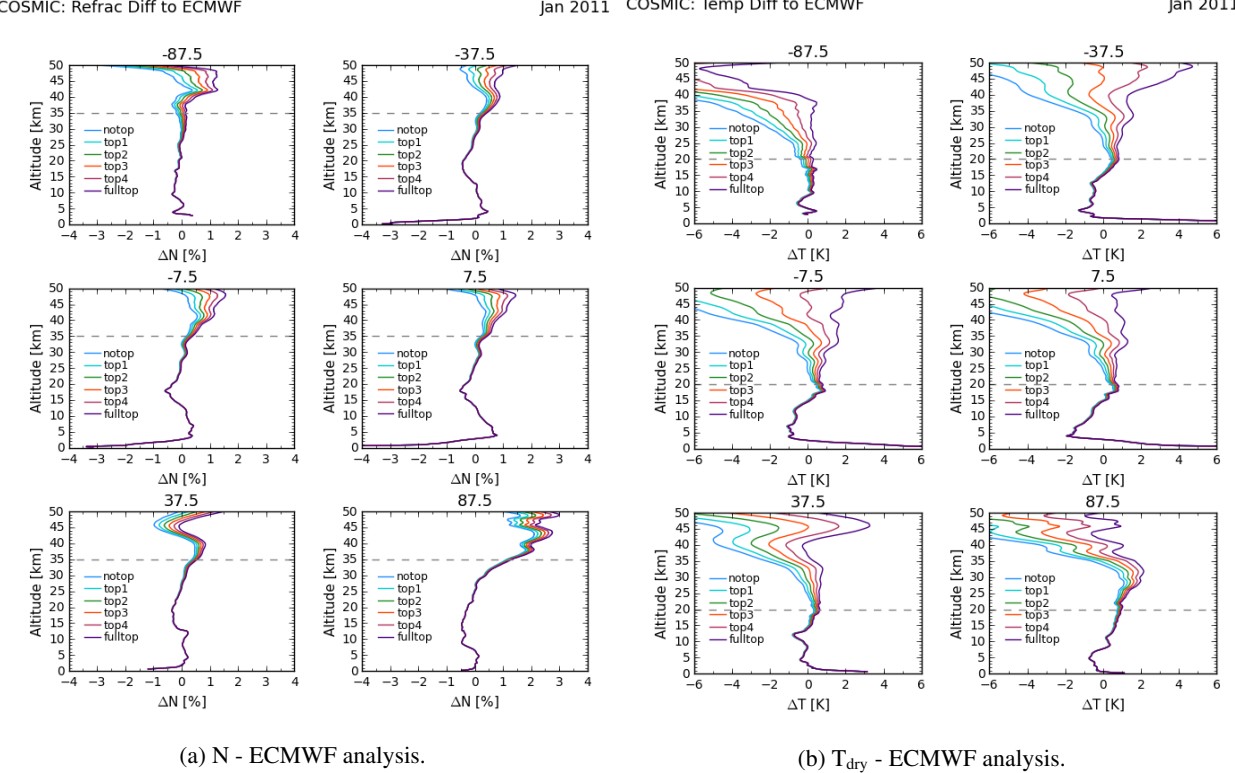

(a) N - ECMWF analysis.

(b) T$_{dry}$ - ECMWF analysis.

**Figure 10.** Sensitivity of single climatological profiles relative to different handling of the altitude above 80 km, analyzed against ECMWF analysis.

Accordingly, Fig. 10b shows dry temperature differences relative to ECMWF for the same mean API climatologies. The plot nicely illustrates the downward propagation of the handling of the top value. At around 20 km altitude differences start to increase between the different choices of initialization, increasing to about ±1 K at 35 km relative to ECMWF analysis. Above 35 km altitude it depends on the choice of the initialization how fast differences increase. The choices of top3, top4 and

5  fulltop seem to agree better relative to ECMWF analysis than notop and small initialization values, such as top1 and top2. This is not surprising, since it is clearly wrong to act as if the bending angle is zero above 80 km. We also compared the different choices of top values amongst each other and also against the choice of exptop. It seems that the values of top3 and top4 are comparable with exptop, i.e., an exponential extrapolation.

## 5.4   API dry temperature climatologies

10  Finally we analyze dry temperature differences relative to the three reference climatologies ECMWF analysis, MIPAS, and SABER. In Fig. 11 we compare WEGC processing and DMI processing, furthermore different choices of the top value are investigated. From top to bottom we analyze WEGC fulltop, WEGC exptop, DMI exptop, and WEGC notop, using as input bending angle climatology DMI L1b data. Starting with the first column, obviously RO API climatologies are in good agree-





ment with ECMWF analysis up to $35\,\mathrm{km}$ altitude. Above that altitude differences start to increase, depending on the choice of the upper initialization. In principle notop makes no physical sense, which is why differences are getting very large relative to ECMWF analysis. The choice exptop leads to very similar results between WEGC (second plot) and DMI (third plot) processing and agrees also up to almost $40\,\mathrm{km}$ altitude very good with ECMWF analysis. Temperature differences vary from $0\,\mathrm{K}$

to about $-1\,\mathrm{K}$. For the choice fulltop, differences are larger (first plot), starting at $20\,\mathrm{km}$ height with about $0.5\,\mathrm{K}$, increasing to about $1\,\mathrm{K}$ at $40\,\mathrm{km}$ altitude. In general, differences to ECMWF analysis tend to be larger at northern high latitudes (USLM disturbance).

Analyzing the dry temperature climatologies in reference to MIPAS data, the general behaviour seems to be relatively similar to ECMWF analysis. WEGC and DMI exptop (second and third plot) agree well up to around $40\,\mathrm{km}$ altitude. Only around

the tropics up to $35\,\mathrm{km}$ altitude small areas of $-0.5\,\mathrm{K}$ up to $-1\,\mathrm{K}$ exist. WEGC fulltop shows stronger differences around the poles compared to MIPAS than compared to ECMWF analysis. On the other hand, SABER data (third column) show much larger differences also in the lower stratosphere up to values of about $-3\,\mathrm{K}$. This is due to a cold bias of SABER data of about $3\,\mathrm{K}$ between $20\,\mathrm{km}$ to $35\,\mathrm{km}$ altitude (Innerkofler, 2015). Furthermore, SABER data show a reduced profile statistics between 90°S to 55°S (about 400 profiles per bin, usually about 1500 profiles per latitude bin), for January 2011. The reduced statistics

is clearly reflected in the SABER plots (third column, Fig. 11).

In Fig. 12 we analyze the differences of the three reference climatologies themselves. We want to understand up to which altitude they show good agreement between each other. Obviously, up to almost $40\,\mathrm{km}$ height ECMWF analysis and MIPAS (first plot) agree very well, although they show little differences of about $\pm0.5\,\mathrm{K}$ in the tropical lower stratosphere and the poles. In the polar region, temperature differences start to increase above $40\,\mathrm{km}$ altitude. On the other hand, SABER exhibits

clearly the cold bias in reference to ECMWF analysis (second plot) and MIPAS (third plot) between $20\,\mathrm{km}$ to $35\,\mathrm{km}$.

Summarized, since ECMWF analysis and MIPAS agree well up to altitudes of about $40\,\mathrm{km}$, they appear to serve as suitable reference climatologies up to this height. Hence, we conclude from our analysis that the exponential extrapolation of WEGC exptop and DMI exptop (second and third row of Fig. 11) is a good choice for the high altitude expansion of the API bending angle climatologies. Data sets between API RO climatologies (WEGC exptop and DMI exptop), ECMWF analysis, and MIPAS

agree very well up to $35\,\mathrm{km}$ altitude and within $\pm0.5\,\mathrm{K}$ to $\pm1\,\mathrm{K}$ at $40\,\mathrm{km}$ altitude.

## 6   Summary and discussion

This work is a follow-up investigation of the so-called average profile inversion (API). The main idea of this method is to propagate average bending angles, instead of individual profiles through the Abel transform. The principle method has been already tested successfully with COSMIC data (Gleisner and Healy, 2013), as well as on CHAMP data (Danzer et al., 2014),

at the Danish Meteorological Institute (DMI). This work here has the focus on a comparison of the new approach between two processing centers, i.e., WEGC and DMI.

We started our analysis with a first attempt to adress the issue of calculating a single mean radius of curvature, $\overline{R_c}$, for a whole bin, although there can be strong variations of $R_c$ from profile to profile. We tested different implementations of mean





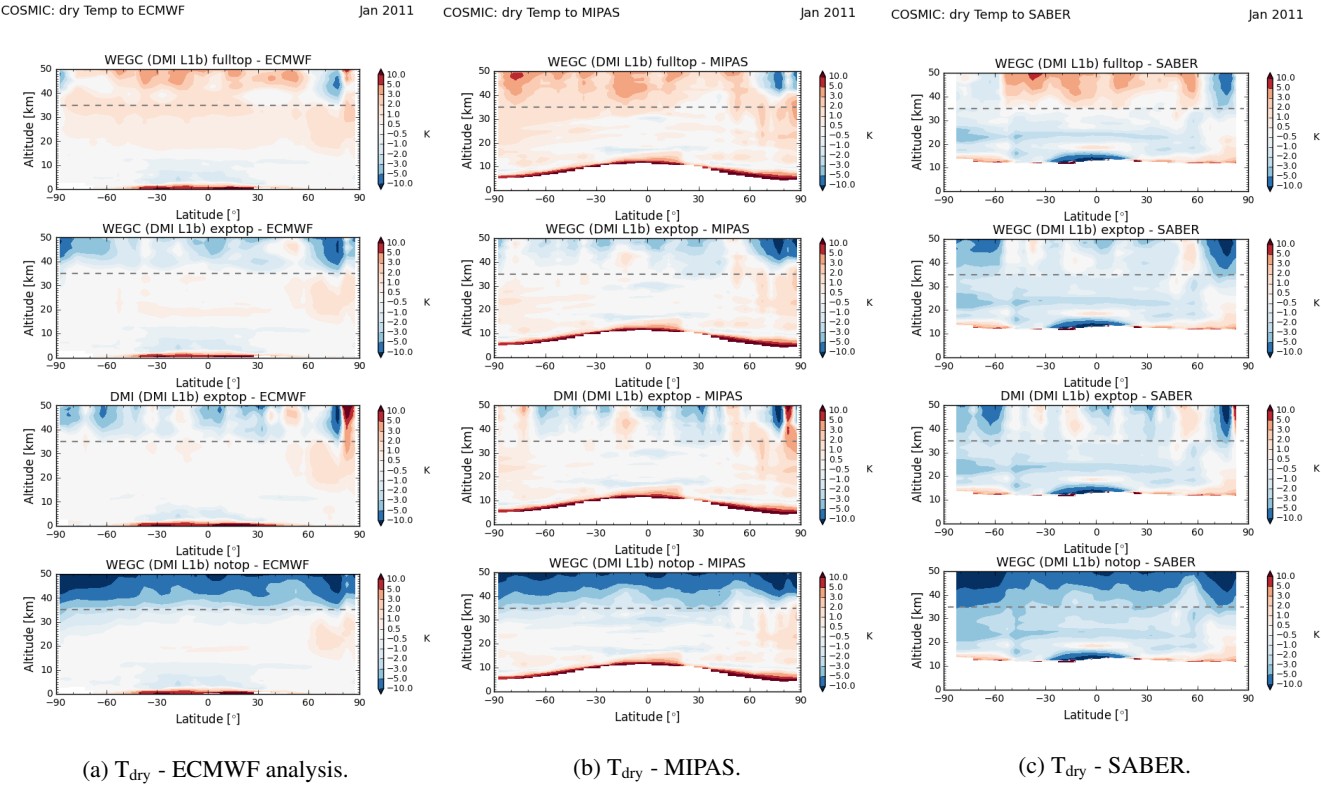

(a) $T_{dry}$ - ECMWF analysis.  (b) $T_{dry}$ - MIPAS.  (c) $T_{dry}$ - SABER.

**Figure 11.** API dry temperature differences relative to different reference climatologies, comparing API processing between WEGC and DMI, studying different high altitude expansions (fulltop, exptop, notop).

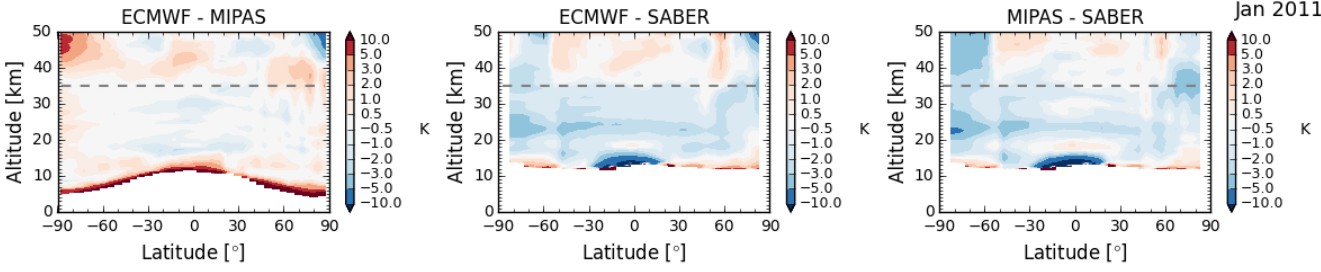

**Figure 12.** Dry temperature differences between reference climatologies.

$\overline{R_c}$ and found that the largest differences are in the tropical area. However, studying the implications of the differences on the RO API dry climatologies, we find negligible impact, which supports the API approach.

Next we tested the API approach in the WEGC processing and compared it to WEGC IPI processing. Although the WEGC rOPS-ex processing system is still in development, we can conclude that differences between the two methods are very small up to $40\,\mathrm{km}$ altitude on refractivity level. Regarding dry temperature climatologies, differences start to exceed $\pm 1\,\mathrm{K}$ above





km height. Hence we conclude that the API method is a valid alternative to the standard inversion for dry atmospheric climatologies up to about 35 km, confirming previous work at refractivity level at the DMI.

For the comparison study between WEGC and DMI we decided to use always the same input bending angle climatologies from DMI, studying monthly 5°-zonal COSMIC data from January until March 2011. That way we can understand differences,
which enter through the different processing systems, and not through the input climatology. The bending angle climatologies are used up to 80 km altitude, above that there is the need for some kind of high altitude expansion due to the Abel integral over infinity. WEGC used monthly ECMWF analysis refractivity fields as top values, while DMI performed an exponential extrapolation with a fixed scale height. Studying the resulting refractivity climatologies, we found that differences between the processing centers start to enter at the altitude of 35 km, see Fig. 5 and Fig. 6. Hence, the choice of the top propagates down
to that respective altitude. This initial analysis showed that the handling of the top is a substantial issue in the average profile inversion.

In a second step we decided to test solely the influence of the different implementations of the Abel integral on our resulting refractivity climatologies. To be able to study that we switched off the high altitude expansion of the average bending angles at both processing centers (notop). Furthermore we tested both, at WEGC and DMI, an exponential extrapolation (exptop).
As a consequence, results became suddenly very similar between WEGC and DMI. For notop the mean refractivities were now almost identical up to 40 km, while for exptop they even agreed up to 50 km. Only in the tropopause region differences of 0.2 % appeared (Fig. 7 and Fig. 8). We conclude that different implementations of the Abel integral do play a minor role, however the handling of the top has a much larger influence.

Next we analyzed the sensitivity of the mean climatologies to the choice of the top value. In that respect especially Fig. 10
is of interest, since it shows the impact of the top value on single mean refractivity climatologies, as well as on the mean dry temperature climatologies. Differences in refractivity start to increase above 35 km altitude, and for dry temperature above 20 km altitude.

Finally, we investigated dry temperature climatologies with respect to the following three different reference data sets: ECMWF analysis, MIPAS, and SABER. Furthermore, we compared different choices of the high altitude expansion (fulltop,
exptop, notop) and also WEGC and DMI processing (see Fig. 11). In general RO API data sets agree well with the reference data sets up to 35 km altitude. For the case of an exponential extrapolation (exptop) they even have a good agreement up to 40 km altitude for both the processing system at WEGC and DMI. Only the fulltop choice leads to enhanced differences starting at about 20 km altitude with 0.5 K, increasing to about 1 K at 40 km altitude.

As a next step we plan to investigate the issue of ionospheric residuals in the bending angle data. For that we will apply the
higher order ionospheric correction method which was introduced by Healy and Culverwell (2015), and further investigated by Danzer et al. (2015); Angling et al. (2017). The correction method is based on the difference of the $L_1/L_2$ bending angles squared and a scaling term $\kappa$, which depends on solar zenith angle, solar flux and altitude. It will be interesting to see if residual ionospheric noise in the data will get reduced - and data quality of the climatologies can be raised to higher altitudes.

In general we conclude that the average profile inversion is a valid, and in respect to computation time even much faster al-
ternative for the production of dry atmospheric RO climatologies. It shows a robustness between the processing centers WEGC





and DMI up to about $35\,\mathrm{km}$ altitude, if different high altitude expansions are used. Applying at both centers an exponential extrapolation, dry temperature climatologies agree between each other, ECMWF analysis and MIPAS climatologies up to $40\,\mathrm{km}$ altitude within $\pm 1\,\mathrm{K}$ .

*Acknowledgements.* We thank UCAR/CDACC for providing COSMIC excess phase data, and the ECMWF for providing analysis data.
5    Furthermore we thank G. Kirchengast for his support and for discussions. Our work was funded by the Austrian Science Fund (FWF) as a
Hertha Firnberg-Project under grant T 757-N29 (NEWCLIM project). H. Gleisner was supported by the ROM SAF, which is a decentralized
operational RO processing center under EUMETSAT. We thank the ROM SAF for providing reprocessed GPS-RO data.





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
