# Peer review of "Comparison study of COSMIC RO dry air climatologies based on average profile inversion"

_Atmospheric Measurement Techniques, 2018_

## Referee Comment (RC1) · Anonymous Referee #1 · 7 Apr 2018

The manuscript "Comparison study of COSMIC RO dry air climatologies based on average profile inversion" by Danzer et al. describes a method to retrieve atmospheric refractivity through averaged bending angles, instead of individual profiles through the Abel transform. And they tested the influence of 'altitude expansion' and 'implementation' by comparing the results with ECMWF reanalysis and satellite data.

I believe the applied analyses methods are sound, but the study only shows the facts but lacks explanations. I think this manuscript needs major restructuring, rewording and general clarifications before the full extent of the presented analyses becomes clear. The study can show some interesting results if they are presented in a precise and detailed manner. I can only recommend the manuscript for publication if major revisions have been done, following the general and detailed comments listed below.

General comments:
1. In the abstract:
Page 1, Line 7:
'Above that altitude some background information for the Abel integral is still necessary. '
Is this a conclusion drawn from this present study? If it is, why it is not explained or discussed in the manuscript at all. The only relevant paragraph is
'The basic idea of the API approach is that averaging of the data in bending angle space suppresses the noise in the data, so that the observed bending angle can be used up to 80 km and the SO step becomes largely obsolete. Above 80km some kind of background information is still necessary. '
in Page 3.
If it is not a conclusion of this study, it is not appropriate mention it in the abstract.

2. The authors compare results with multiple data, namely the reanalysis data , and satellite data MIPAS and SABER
I have some questions here:
1) If there is a very good agreement between the MIPAS and SABER temperature data, as you mentioned in Page 7 Line 8, what is the point to compare your results with both of them?
2) Each satellite instrument has its own sensitive altitude range and accuracy. Have you consider the accuracy of the satellite data themselves?
3) You may also need to talk about the horizontal resolution of these data and its potential influence on the comparison.

3. Clearly your inversion results vary with latitudes, but does the accuracy of your inversion result vary with seasons? And will your inversion results influenced by humidity? Although it is the 'dry temperature' you are studying, water vapor in the atmosphere may significant influence the excess phase, right?

4. Please try to explain why the largest differences are around 35 km in fig. 5-7, 9-10.
5. Why there are large differences in tropics and mid-latitudes near surface in fig. 5-7,9 and how does the inversion from negative to positive differences formed, e.g. at ~2-3km in the tropics in fig.5

6. All your results are based on COSMIC excess phase from Jan to Mar 2011. So I guess if your results depend on seasons, your conclusions are only valid in January to March. Please refine the way that you describe your conclusion.

7. In Sect. 6 Summary and discussion, the authors summarized the study and talked about the outlook of the study. I would say Sect. 6 is only a summary but not a decent discussion at all.
   In fact, in the whole manuscript, the authors have made a very comprehensive comparison, but they focused only on the 'fact' but ignored the 'reason'.
   I suggest the authors add a separate section of discussion before the summary, in which all the problems and uncertainties of the present study should be discussed in a more detailed manner.
   And in the section of summary and/or conclusion, the authors should show readers very clear the conclusion from this present study, not from previous study or future work.

Specific comments:
Page 1, the format of the superscript number in the authors' names might not correct.
   Line 9 follow-up

Page 2,
   Line 4 delete meanwhile;
   numerical weather prediction (NWP) and climate monitoring in the upper troposphere and lower stratosphere (UTLS) (however, I believe the GPS RO data do not only valuable in the UTLS but in both troposphere and stratosphere, and one or more references are needed here.)
   Line 13 Please explain the abbreviations uniformly in the manuscript, COSMIC (Constellation Observing System for Meteorology, Ionosphere, and Climate) or COSMIC Data Analysis and Archiving Center (CDAAC).
   The abbreviations have to be explained, but ONLY at the FIRST time when they are used. Please fix the abbreviations in the manuscript.
   Line 15 Foelsche et al. 2011 reference is missing.
   Line 16 'level, but' → 'level but'

   Line 2. Please pay attention to the requirements of AMT regarding the 'Section/Sect.' and 'Figure/Fig.' and correct them through the whole manuscript.
   Line 8. I prefer 'Abel transform'
   Line 27 'mean-sea level altitude' → 'mean sea level'

Line 30 'Kirchengast et al., 8-14 September 2016, 2017', remove '8-14 September 2016,'

Line 6 'As a next step the refractivity is calculated, applying the method described in Appendix B in Syndergaard and Kirchengast (2016) on the residual state'. I can understand that the refractivity is calculated in the next step by applying the method described in … But this sentence is too strange to me. Please rephrase it.

Line 10 'Sec.' → 'Sect.'

Line 24 ' 5 °-zonal': difficult to understand. Do you indicate mean value over 5 degrees in latitude? Please make it clear.

Line 28 '1/1000K', '1/100K'? Do you mean '0.001K' and '0.01 K'? Please change them.

Line 4 '1/5 incremental steps'. I can't understand.

Line 15 individual

Line 1 'hence' → 'so', or ',' → ';'

Line 21 'Summarized,' → 'To summarize,'

Line 32, address

Figure 1:

Left panel: what does the blue dashed line indicate? Please explain.

I would strongly recommend that the authors find a native English speaker to check the manuscript for grammar and structural problems.

The authors should also fix typos that I did not list (because I am out of patience…).

The authors should first read the 'manuscript preparation guidelines for authors' (https://www.atmospheric-measurement-techniques.net/for_authors/manuscript_preparation.html) and refine the manuscript before submitting it.

---

## Referee Comment (RC2) · Anonymous Referee #2 · 1 May 2018

Review of "Comparison study of COSMIC RO dry air climatologies based on average profile inversion" This study applied the API method to retrieve refractivity and dry temperature climatology for COSMIC data from January to March 2011. The method is not new, but a thorough comparison against multiple data could be very interesting. However, the paper is not well organized, and the presentation cannot be followed smoothly. The structure of the paper needs to be refined. Furthermore, some results lack of insightful explanation, or have no interpretation at all. The writing is another major issue. There are many grammar mistakes and typos. To make this work publishable, the authors should consider a thorough revision.

P: page, L: line Here are some of my major concerns, • The abstract could be rewritten with major points of conclusion from this study. • check grammar and language

[Figure]

• re-structure and consider the way of presenting. For instance, the method of API may be presented immediately after the first sentence. • L17. The authors use different terms, e.g., upper initialization, upper boundary value, and top. They need to be clear, precise and consistent. • P2L4, is that only in UTLS? Why? • P2L30, "up to high altitudes", how high is it? "introduced an alternative approach", I guess it is not an alternative approach, but a different application? Please clarify. • What is the major benefit of the API method? While it is comparable to IPI below 35 km, I see it is not very helpful in extending the accuracy of retrieval above 35km. Is it computational efficient? If so, can the authors provide the computational cost of the API and IPI? • "The averaging of a large number of profiles suppresses noise in the data, enabling observed bending angle data to be used up to 80 km without the need of a priori information." I do not understand. Can the authors explain more on this? which figures or results support this point and how? I did not see the connection of the current results to benefit of using bending angle data between 35 and 80 km. • Definitions of M and N in Equation 3 do not seem correct. • Many figures and results lack of complete explanation. I just list some of them as below, • Figure 1, "only negligible implications are found". Why are the dry temperatures retrieved using different Rc identical? What does "implications" mean? What is the reason for the large differences between 2–8 km? • Figure 2, please explicitly provide what the dashed straight lines are. I think impact height is more accurate than impact altitude? • Figure 3, what is the reason for the greater than 0.8 % difference around tropopause in refractivity? What is the reason for the large differences in the lower atmosphere (near surface)? What does altitude mean in the y axis? Is it impact height? How is the percentage calculated? Is the difference normalized by something? • Figure 4, there is no description at all. What is the purpose of putting this figure? • Figure 5, what does "data show again a slight increase" mean? What increases? Again, what is the explanation for the near surface differences? • Figure 9, the authors could provide more explanation for the large differences in the northern high latitudes. • Figure 10, "increasing" to about +/- K is not accurate. It seems the patterns among the choices are different for the

bins in the northern/southern hemisphere. Are the results showing here season dependent? • Summary and discussion: Instead of repeating the major steps of what was already presented, the authors need to highlight the major points, and discuss the limitation and generalization of this study.

Some Minor comments (there are many more), • P1L3, remove already • P1L3, propagating –> propagate • P1L17, what does already mean? • P1L14, expansion –> extension, check all usages throughout the paper • P1L19, between –> among • P1L20, P13L9, average profile inversion –> API • P2L2, remove meanwhile • P2L5, global analyses and forecasts? • P2L6, simulations might be simulation? • P2L8-L9, "NWP centers will always assimilate data that are as close as possible to the original measurement; in case of RO these are atmospheric bending angles, which can be assimilated without any bias correction." What do the authors mean by will? and what does "these" mean? • P2L16-L17, please check grammar, "which is still small at bending angle level, but increase through the retrieval chain." • P2L24, change "Ao et al. (2012); Gleisner and Healy (2013)" to Ao et al. (2012); and Gleisner and Healy (2013). Also see P18L31. • P3L1, "at WEGC and DMI," –> "at WEGC and DMI." • P3L2, please rewrite "section 3 ….." • P3L8, "Abel transformation" or "Abel transform"? • P3L30, please correct the format of the citation. • P4L3, change "(2017a)" to 2017a • P4L9, change "(2017b)" to 2017b; also check P10L5 • P4L11, "up to", I think "below" is easier to understand. Also check other usages. • P4L23 and Figure 1, I recommend using left/right or putting (a)/(b) in the figure instead of using l.h.s./r.h.s; what are the two dashed blue curves? It seems they are not mentioned in the paper. • P4L24, what do the authors mean by 5O-zonal? Please be clear and precise. • CDAAC and UCAR should be used (formatted) consistently throughout the paper, e.g., P6L11, P5L11, etc. • P6L12, Level L2a processing? • P6L15, IPI was already introduced on P3. • P6L24, please pay attention to and check the use of hence. • P6L29, use full term at the first time and use abbreviation for the rest consistently. Please also check other usages. • P6l30, again, what is "5o latitudinal bins"? • P6l30, in –> at •

P7L1-L2, remove content in brackets since they were already given in P3. • P8L5, L10, please be consistent on rOPS-ex and rops-ex • P8L21, what does the "RO core region of 35 km" mean? • P10L14, alternative • P19L4, CDACC–> CDAAC

Please also note the supplement to this comment:
https://www.atmos-meas-tech-discuss.net/amt-2018-29/amt-2018-29-RC2-supplement.pdf

---

## Author Comment (AC1) · 8 Jun 2018

**Author's Response to Referee #1**

We would like to thank referee #1 for the thorough evaluation of our manuscript. We have answered all comments below (for easier comparison the referee comments are included in *italic*).

General comments:

*#1: In the abstract:*

*Page 1, Line 7:*

*'Above that altitude some background information for the Abel integral is still necessary.' Is this a conclusion drawn from this present study? If it is, why it is not explained or discussed in the manuscript at all. The only relevant paragraph is*

*'The basic idea of the API approach is that averaging of the data in bending angle space suppresses the noise in the data, so that the observed bending angle can be used up to 80 km and the SO step becomes largely obsolete. Above 80km some kind of background information is still necessary. ' in Page 3. If it is not a conclusion of this study, it is not appropriate mention it in the abstract.*

**1:We tried to keep the abstract as concise as possible, but we agree with the referee that some additional information is needed for context. Therefore, we added at the top of the abstract:**

Global Navigation Satellite System (GNSS) Radio Occultation (RO) data allow for the retrieval of near vertical profiles of atmospheric parameters like bending angle, refractivity, pressure and temperature. The retrieval step from bending angle to refractivity, however, involves an Abel integral, whose upper limit is infinity. RO data are practically limited to altitudes below about 80 km and the observed bending angle profiles show decreasing signal-to-noise ratio with increasing altitude. Some kind of high-altitude background data are therefore needed, in order to perform this retrieval step (this approach is known as "high-altitude initialization"). Any bias in the background data will affect all RO data products beyond bending angle. A reduction of the influence of the background is therefore desirable – in particular for climate applications. Recently, …

Furthermore we will add on p.3, line 21/22:

Above 80 km the bending angle still needs to be extended, since the Abel integral is over infinity and the bending angle is not zero above 80 km. Different extensions of the bending angle are tested in this study, see description in Sect. 2.1 and Sect. 2.2

We will also add the following citation on p. 3, line 27:

(for details see Gleisner and Healy, 2013; Danzer et al. 2014).

*#2: The authors compare results with multiple data, namely the reanalysis data , and satellite data MIPAS and SABER*

*I have some questions here:*

*1) If there is a very good agreement between the MIPAS and SABER temperature data, as you mentioned in Page 7 Line 8, what is the point to compare your results with both of them?*

*2) Each satellite instrument has its own sensitive altitude range and accuracy. Have you consider the accuracy of the satellite data themselves?*

*3) You may also need to talk about the horizontal resolution of these data and its potential influence on the comparison.*

**#2:** MIPAS and SABER provide independent measurements of the atmosphere, using different retrievals. Therefore we are convinced that it is useful to compare RO data with both data sets. Even if data sets are in good agreement it is valuable to see if the new data set is in line with the reference data sets. MIPAS and SABER data sets show high accuracy results in the stratosphere, and are hence interesting for a dry atmosphere comparison study. The second paragraph on p.7, lines 10-15 gives an overview of how MIPAS and SABER temperature data sets compare relative to WEGC RO data using a standard processing (Innerkofler, 2015). We think it is interesting to see if similar results are achieved relative to those reference data sets when comparing RO API data sets with different high altitude expansions, instead of RO IPI data sets. However, we see that this has to be put in a better context.

We will extend the discussion and include another paragraph in Sect. 6, p.18, after line 28:

The temperature comparison study of RO API data sets relative to ECMWF analysis, MIPAS, and SABER data sets shows for both, the WEGC and the DMI exptop case, similar temperature biases as Innerkofler (2015) found in his study analyzing global RO IPI temperature data sets. RO API, ECMWF analysis data, and MIPAS data agree within +-1K, up to 40 km. Above 40 km they begin to show larger differences than when analyzing global

RO IPI data. Furthermore, the 3K temperature bias of SABER data could also clearly be illustrated relative to RO API data.

Concerning the horizontal resolution:

It is true that the underlying observational data sets have different horizontal resolutions. However, these data are not directly compared. Comparisons are made between data that have been averaged in monthly latitude bins, which are identical for the RO, ECMWF, MIPAS, and SABER data sets. The different horizontal resolutions of the underlying data is not a major problem for these heavily averaged data sets.

*#3:  Clearly your inversion results vary with latitudes, but does the accuracy of your inversion result vary with seasons? And will your inversion results influenced by humidity? Although it is the 'dry temperature' you are studying, water vapor in the atmosphere may significant influence the excess phase, right?*

**3: The influence of humidity is important in the troposphere and has also been studied by Danzer et al. (2014). For the present study, where we analyze the influence of the high altitude initialization - which is important in the stratosphere, the influence of humidity is negligible. We also do not look at seasonal dependence of the API method. However, a long term study of the API method has already been performed in a previous work of us with CHAMP data (Danzer et al., 2014), where data sets from September 2002 until September 2008 have been analyzed. The study did not indicate that the accuracy of the inversion itself does depend on season. However, the study showed that differences relative to reference data sets increase towards higher latitudes, for both, the API and IPI inversions. We focus in this study on the three COSMIC test months January to March 2011, since this work is a follow-up investigation of Gleisner and Healy (2013), who also tested the same three months at the DMI .**

*#4: Please try to explain why the largest differences are around 35 km in fig. 5-7, 9-10.*

**4: We are not completely sure if this remark refers to the general increase in differences beyond 35 km altitude (Fig. 5, 6) or to the larger differences relative to ECMWF analysis at high northern latitudes (Fig. 7,9), therefore we tried to answer both:**

1) The "core region" of RO data is between 5 km to 35 km, hence the dashed line in the figures is always plotted at 35 km. The high accuracy in this region has been shown in previous studies, such as Steiner et al. (2013), who showed that consistency between different sets from different processing centers is highest in the UTLS. Hence, regarding the API approach, it is not surprising that differences also start to increase above that respective

altitude. We emphasize that the RO – ECMWF biases above 35 km that we see here are not related to the API method. They are generally seen in all RO - ECMWF comparisons (see also Figures 5,6,7 in Gleisner and Healy (2013) which compares API and IPI relative to ECMWF analysis).

However, we see that it is necessary to emphasize this more strongly in the manuscript and we will discuss this in Sect. 6 Summary and discussion. According to your suggestion, we will rewrite this section and extend the discussion part.

On page 2, line 4 we will add:

The altitude range from 5 km to 35 km is therefore commonly regarded as the "core region" of the RO technique.

Furthermore, on p. 18, line 9.

… The observed RO – ECMWF biases above 35 km are not related to the API method. They are generally seen in all RO - ECMWF comparisons when applying the standard processing (see comparison of API and IPI relative to ECMWF analysis in Figures 5,6,7 in Gleisner and Healy (2013)). In that context it is interesting to see that different handling of the top value above 80 km also propagates down to that respective altitude. ...

On p. 18, after line 22 the following paragraph:

Steiner et al. (2013) showed in a comparison study of climate data products from six international processing centers that different high altitude initialization approaches affect uncertainties in CHAMP RO data from about 25 km upwards. Largest differences between processing centers are found towards increasing altitudes and at high latitudes. This has also been demonstrated for the API approach in a prior study analyzing CHAMP data (Danzer et al., 2014), where differences relative to ECMWF analysis also increased towards high altitudes and latitudes. Also the API approach shows an increasing sensitivity above 35 km altitude when comparing different high altitude expansions for the bending angle, as well as, comparing WEGC and DMI processing centers. The illustrated propagation of uncertainties downwards through the API retrieval chain to about 20 km in dry temperature has also been observed in prior studies for standard retrievals from different processing centers (Foelsche et al., 2011; Ho et al., 2012; and Steiner et al., 2013).

2) On p.14, lines 3-5, we will add another paragraph, including two reference:

Differences relative to ECMWF analyses are lager at northern high latitudes, which could be related to different sampling of the upper stratosphere lower mesosphere (USLM) disturbance in January 2011 (Greer et al., 2013). Related to that, the Arctic winter 2010/2011 has been

notified as one of the coldest stratospheric winters on record (Sinnhuber et al., 2011).

*#5: Why there are large differences in tropics and mid-latitudes near surface in fig. 5-7,9 and how does the inversion from negative to positive differences formed, e.g. at ~2-3km in the tropics in fig.5*

**5: The focus of the study is the stratosphere, where the API method has decisive advantages in comparison with the IPI method. The main purpose of figures 5-7 and 9 are to show the impact on the stratospheric refractivity retrievals by different factors, such as, DMI/WEGC differences and different high-altitude expansions. The refractivity bias structure in the low- and mid-latitude troposphere in the lowest few kilometers seen in figures 5-7 and 9 is not caused by the API method. The bias structure is well-known and is also seen in the IPI method relative to ECMWF analysis. Please see Figures 5,6,7 in Gleisner and Healy (2013). However, the error at the lowest ~2 km is probably due to the use of a mean radius of curvature. This error can also seen in the comparison of API to IPI in Figure 4 of Gleisner and Healy (2013).**

We will therefore add (page 10, line 15):

Please note that the focus of this study is the stratosphere and that we therefore show dry parameters, which are not fully adequate to characterize moist regions in the lower troposphere. The refractivity bias structure in the low- and mid-latitude troposphere in the lowest few kilometers relative to ECMWF is not caused by the API method. It can also be seen for the IPI method (see Figures 5,6,7 in Gleisner and Healy (2013)). However, the error at the lowest ~2 km is probably due to the use of a mean radius of curvature.

*#6: All your results are based on COSMIC excess phase from Jan to Mar 2011. So I guess if your results depend on seasons, your conclusions are only valid in January to March. Please refine the way that you describe your conclusion.*

**6: Please see answer #3. Furthermore, for clarifications we will include the following sentences in Sect. 6:**

p.2, line 33

In this study, we test different implementations of the API approach at the Danish Meteorological Institute (DMI) and the Wegener Center for Climate and Global Change (WEGC) and validate them against independent data. We analyze three COSMIC test months

from January to March 2011, following the investigations of Gleisner and Healy (2013). A long term API data set study has already been performed for the complete CHAMP period (Danzer et al., 2014), and is not part of this investigation.

*#7: In Sect. 6 Summary and discussion, the authors summarized the study and talked about the outlook of the study. I would say Sect. 6 is only a summary but not a decent discussion at all. In fact, in the whole manuscript, the authors have made a very comprehensive comparison, but they focused only on the 'fact' but ignored the 'reason'. I suggest the authors add a separate section of discussion before the summary, in which all the problems and uncertainties of the present study should be discussed in a more detailed manner. And in the section of summary and/or conclusion, the authors should show readers very clear the conclusion from this present study, not from previous study or future work.*

**7: According to your suggestion we will rewrite Sect. 6 and extend the discussion part. Furthermore we will rename Sect. 6 to "Summary, discussion and outlook"**

Specific comments:

We do not list the complete number of specific comments. However, we thank the referee for the thorough reading of the manuscript and will perform the necessary changes according to your suggestions.

Only specific comments, which require an answer, are listed here:

**1: *Page 2, line 4**

*numerical weather prediction (NWP) and climate monitoring in the upper troposphere and lower stratosphere (UTLS) (however, I believe the GPS RO data do not only valuable in the UTLS but in both troposphere and stratosphere, and one or more references are needed here.)*

**1: We thank the referee for his valuable comment about the utility of RO data: You are right, but the highest quality (and the highest impact on NWP analyses) is clearly achieved in the UTLS. We will change the first sentence of the introduction to:**

... Monitoring, *in particular* in the Upper Troposphere and Lower Stratosphere (UTLS).

The general goal is to expand this altitude range and to increase the utility of RO data (towards the bottom, as well as towards increasing altitude). This study attempts to increase

the utility in the (upper) stratosphere.

The citations are given in the same paragraph in the next three lines (p.2, lines 5-7), first referring to NWP, then to Climate.

We will add to the introduction on p. 2, after line 31.

The advantages of the API approach are the following, a) the reduction of background in the data, b) the circumvention of the complicated statistical optimization step (a known reason for differences between processing centers), c) the API approach is much faster in computation.

Furthermore we extend the paragraph on p.2, line 33

...The aim of the API approach is to produce high quality climatologies, with well characterized errors, which might push current limits in altitude further, enabling the study of stratospheric climatologies above 35 km.

In the discussion on p. 19, line 3 we add the following sentences:

The latter result might suggests that API dry temperature climatologies can be used up to 40 km, pushing current limits of the utility of RO data in the stratosphere.

*#2: Figure 1: Left panel: what does the blue dashed line indicate? Please explain.*

**2: Thank you for noticing. It is the standard deviation of AvProf. We will write:**

p.4, line 26

(Eq. 2, AvProf – blue line, its standard deviation  - blue dashed line)

*#3: I would strongly recommend that the authors find a native English speaker to check the manuscript for grammar and structural problems.*

**3: We will follow your suggestion and have asked a native speaker to perform final proof-reading on the revised manuscript.**

---

## Author Comment (AC2) · 26 Jun 2018

**Author's Response to Referee #2**

We would like to thank referee #2 for the thorough evaluation of our manuscript. We have answered all comments below (for easier comparison the referee comments are included in *italic*).

General comments:

*#1: The abstract could be rewritten with major points of conclusion from this study.*

- *check grammar and language*

- *re-structure and consider the way of presenting. For instance, the method of API may be presented immediately after the first sentence.*

- *L17. The authors use different terms, e.g., upper initialization, upper boundary value, and top. They need to be clear, precise and consistent.*

**1: Related to the comment of referee #1 we have already restructured our abstract in order to clarify open questions. We invite you to read answer #1 to referee #1, this should also help with some of your concerns.**

Regarding different terms: Thank you very much for your input. We will limit the number of terms by replacing "upper boundary value" with "top value", and "upper initialization" with "high altitude initialization" throughout the entire manuscript.

*#2: P2L4, is that only in UTLS? Why?*

**2: The core region of RO data is the UTLS. Studies show highest consistency between different data sets in that respective altitude range, see e.g., Steiner et al. (2013). The reasons are ionospheric residuals and a decreasing signal-to-noise ratio with increasing altitude (see e.g., Danzer et al., 2013). In the lower troposphere (below 7 km) – which is not the focus of this study - the error budget is dominated by horizontal variations of refractivity, and consequent deviations from the spherical symmetry assumption (e.g., Healy, 2001). The data can be affected by signal multi-path and super-refraction, and the temperature retrieval requires background information (e.g., Sokolovskiy et al., 2010).**

We invite you to read a more detailed answer in our response to referee #1, question #4, and in related citations, given e.g., on p2/l17. Furthermore, we also intend to add further

information in our manuscript (see also question #4/referee #1).

*#3: P2L30, "up to high altitudes", how high is it? "introduced an alternative approach", I guess it is not an alternative approach, but a different application? Please clarify.*

**3: The BAROCLIM spectral model reaches formally up to infinity. The idea of the model is to use the average bending angles (which are also combined at altitudes above about 60 km with the MSIS-90 climatology) as a priori information in the statistical optimization step of the processing of individual bending angle profiles. Details of the BAROCLIM spectral model are given by Scherllin-Pirscher et al., 2015. At the DMI the model has been implemented as their background climatology in the new ropp processing system. The difference to our approach is that BAROCLIM serves as a background climatology for the statistical optimization step of individual bending angles, while we avoid statistical optimization completely and process climatologies.**

*#4: What is the major benefit of the API method? While it is comparable to IPI below 35 km, I see it is not very helpful in extending the accuracy of retrieval above 35km. Is it computational efficient? If so, can the authors provide the computational cost of the API and IPI?*

**4: The major benefit is that bending angles are used up to 80 km altitude instead to about 35 km altitude, when statistical optimization is applied. The aim is always to use less background in the data, and the hope is - with less background, that the utility of the climatologies can be pushed above 35 km. Furthermore it is much faster, e.g., the difference to processing 500 profiles or just one profile. See also specific comment #1 to referee #1, where we stated to add:**

Introduction on p. 2, after line 31.

The advantages of the API approach are the following, a) the reduction of background in the data, b) the circumvention of the complicated statistical optimization step (a known reason for differences between processing centers), c) the API approach is much faster in computation.

Furthermore we extend the paragraph on p.2, line 33

...The aim of the API approach is to produce high quality climatologies, with well characterized errors, which might push current limits in altitude further, enabling the study of stratospheric climatologies above 35 km.

In the discussion on p. 19, line 3 we add the following sentences:

The latter result might suggests that API dry temperature climatologies can be used up to 40

km, pushing current limits of the utility of RO data in the stratosphere.

*#5: "The averaging of a large number of profiles suppresses noise in the data, enabling observed bending angle data to be used up to 80 km without the need of a priori information." I do not understand. Can the authors explain more on this? which figures or results support this point and how? I did not see the connection of the current results to benefit of using bending angle data between 35 and 80 km.*

**5: The averaging of the data leads to a rather smooth mean bending angle profile up to an altitude of 80 km, compared to the noisy individual profiles, which suffer with increasing altitudes from increasing problems with measurement noise and also ionospheric residuals. This manuscript is not a proof of concept paper. It is a follow up comparison investigation, focusing on the comparison between two processing centers. For better context we added an additional paragraph in the abstract introducing the problem of RO data at high altitudes (see answer #1 to referee #1). For the basic introduction and analysis of the method please see Gleisner and Healy (2013), and also the paper about the application to CHAMP data, Danzer et al. (2014). Regarding the benefits, please see answer #4.**

*#6: Definitions of M and N in Equation 3 do not seem correct.*

**6: Thank you very much for noticing! In the definition of M is a mistake in the numerator. It is $(ab)^2$ and not $ab^2$. We will correct it immediately.**

*#7: Many figures and results lack of complete explanation. I just list some of them as below,*

- *a) Figure 1, "only negligible implications are found". Why are the dry temperatures retrieved using different Rc identical? What does "implications" mean? What is the reason for the large differences between 2–8 km?*

- *b) Figure 2, please explicitly provide what the dashed straight lines are. I think impact height is more accurate than impact altitude?*

- c) *Figure 3, what is the reason for the greater than 0.8 % difference around tropopause in refractivity? What is the reason for the large differences in the lower atmosphere (near surface)? What does altitude mean in the y axis? Is it impact height? How is the percentage calculated? Is the difference normalized by something?*

d) *Figure 4, there is no description at all. What is the purpose of putting this figure?*

e) *Figure 5, what does "data show again a slight increase" mean? What increases? Again, what is the explanation for the near surface differences? Figure 9, the authors could provide more explanation for the large differences in the northern high latitudes.*

f) *Figure 10, "increasing" to about +/- K is not accurate. It seems the patterns among the choices are different for the bins in the northern/southern hemisphere. Are the results showing here season dependent?*

**7:**

a) The local radius of curvature (Rc) can be illustrated in two extreme ways. On the one hand as "local radius of curvature in north-south (meridian) direction, i.e., $M(\varphi)$" and on the other hand as "local radius of curvature in east-west (normal to meridian) direction, i.e., $N(\varphi)$". Their largest difference is at the equator, while at the poles they are equal:

See also https://en.wikipedia.org/wiki/Earth_radius#/media/File:EarthEllipRadii.jpg

We will write on p. 4, line 26:

"… differences increase in the tropics between about 2 km and 8 km. The reason is that the local radius of curvature in north-south (meridian) direction, i.e., $M(\varphi)$, and the local radius of curvature in east-west (normal to meridian) direction, i.e., $N(\varphi)$, show maximum differences at the equator, while at the poles they are equal. When building a mean Rc, $M(\varphi)$ and $N(\varphi)$ were either averaged by using the Mean or the Gaussian formula (Eq. 3 and Eq. 4). In case of a single RO measurement the radius of curvature is a result of the momentary orbit geometry of the two involved satellites (GNSS and LEO). Using as a third formulation a simple averaging of all radii of curvature in a bin, we therefore find the largest differences between ± 30 degrees latitude (see l.h.s. Fig. 1). However, the impact of the different formulations of Rc on dry temperature was found to be negligible in the stratosphere, see r.h.s. of Fig. 1**.** The variations are between about … "

b) The dashed lines at 50 km and 60 km are simply a help to mark the transition region of the medmean bending angles. We will include this in the text for clarifications.

The difference "Impact Height" to "Impact Altitude": Impact Height is the height above the ellipsoid, using the WGS-84 model. Impact Altitude is the height above the geoid (see Scherllin-Pirscher et al., 2017). One altitude is not more accurate than the other.

c) In this study the focus is the stratosphere, and hence, we only discuss dry parameters. It is however a very valid question which we also answered in question #5 to referee #1.

The altitude in the y-Axis means altitude above the geoid.

Yes, the percentage is normalized. The figures show for refractivity the relative difference, as in the primary paper Gleisner and Healy (2013). Thank you very much for this comment. We will add a sentence to the paper on p. 8, after the sentence from line 10:

All refractivity differences are studied as relative differences (given in percentage), while the temperature differences are studied in absolute differences (given in Kelvin).

d) The description is given on p.9, line 6, continuing to p.10, lines 1-2. The plot shows the mean bending angle profiles of the DMI relative to ECMWF analysis for January 2011.

e) Regarding the sentence "data show again a slight increase" we have to apologize. The word "again" needs to be deleted → "data show a slight increase relative to ...".

Concerning the near surface differences, please see answer #5 to referee #1.

The large northern high latitude differences are related to an upper stratosphere lower mesosphere (USLM) disturbance in January 2011 (Greer et al., 2013) and a very cold stratospheric Arctic winter in 2010/2011. Please see answer 4 to referee #1.

f) Thank you very much. We will rewrite the sentence in the following way:

"... increasing to about a 2-3 K difference at 35 km altitude relative to .."

No, the results are not season dependent (see also in more detail the answers #3 and #6 to referee #1). The large northern high latitude differences are due to the very cold stratospheric Arctic winter.

*#8: Summary and discussion: Instead of repeating the major steps of what was already presented, the authors need to highlight the major points, and discuss the limitation and generalization of this study.*

**8: We will follow the suggestion of you and referee #1. Large parts of the summary will be rewritten. Parts of the revised text are already specifically written down in our answers to you and referee #1.**

Minor comments:

We do not list the complete number of minor comments. However, we thank the referee for the thorough reading of the manuscript and will perform the necessary changes according to

your suggestions.

Only minor comments, which require an answer, are listed here:

*#1: P2L8-L9, "NWP centers **will** always assimilate data that are as close as possible to the original measurement; in case of RO **these** are atmospheric bending angles, which can be assimilated without any bias correction." What do the authors mean by will? and what does "these" mean?*

**1: We will rephrase the sentence in the following way:**

"At most NWP centers, RO data are assimilated in the form of bending angles, not in the form of geophysical variables retrieved from the bending angles. Climate monitoring based on RO data, on the other hand, requires the full range of geophysical parameters, from refractivity ....."

*#2: P4L24, what do the authors mean by 5° -zonal? Please be clear and precise.*

**2: Monthly 5°-zonal COSMIC data means all data of the COSMIC mission from one month, averaged in 5°x360° latitude x longitude steps.**

*#3: P8L21, what does the "RO core region of 35 km" mean?*

**3: The RO core region of 35 km is the region between 5 km to 35 km, where highest data quality is found. See also answer #2. Clarifications will be included in the revised manuscript.**